# Effect of vacuum freeze drying and hot air drying on dried mulberry fruit quality

Li Wang[1,2], Haichao Wen[1,2], Ningwei Yang[1,2], Hongjiao Li[1,2]*

1 College of Forestry, Agricultural University of Hebei, Hebei, P.R. China, 2 Key Laboratory of Germplasm Resources of Forest Tree and Forest Protection of Hebei Province, Baoding, China

* lihongjiao0103@163.com

**Data Availability Statement:** All relevant data are within the paper and its Supporting information files.

**Funding:** Hongjiao Li, Natural Science Foundation of Hebei Province (C2019204257) Youth Fund for

## Abstract

Two different drying methods (vacuum freeze-drying and hot-air drying) were used to dry mulberry of three varieties 'Baiyuwang'(D1), 'Longsang'(D2) and 'Zhongshen.1'(D3), and the fresh fruit of each variety was used as the control. The effects of different processing conditions on the physical characteristics, nutrients, functional components and antioxidant activity of mulberry fruit were analyzed. The results show that after different drying methods, after vacuum freeze-drying, the physical properties of dried mulberry fruit such as wettability, hygroscopic property and water retention, soluble protein, ascorbic acid and other nutrients, functional components such as polyphenols, resveratrol, chlorogenic acid and anthocyanin, and antioxidant activities such as DPPH free radical scavenging ability and ABTS free radical scavenging ability were superior to hot air drying (P < 0.01). It was concluded that vacuum freeze drying was more beneficial for retaining the original quality of mulberry than hot air drying. This study can provide a retaining theoretical basis for mulberry deep processing and comprehensive development and utilization.

## 1 Introduction

Mulberry, also known as mulberry fruit, mulberry jujube, black mulberry, etc., is the mature ear of mulberry (Morus alba L.). Mulberry has high nutritional value and is rich in amino acids [1], proteins [2], vitamins [3], minerals and other nutritional elements. It is also rich in polysaccharides [4, 5], alkaloids [6], flavonoids, resveratrol [7], anthocyanins [8] and other functional active ingredients, which have the medicinal value of enhancing immunity, antioxidants, regulating blood lipids and lowering blood sugar [9–11]. It is known as "the third generation of new fruit", and by the Ministry of Health as a drug and food dual-use plant resource [12].

Mulberry is a berry, that is tender and juicy, sweet and sour, and most mature in late spring and early summer with high temperature and high humidity. Due to the high water content of its fruits, it brings great difficulties in storage, transportation and preservation. In addition, it is easy to damage and rot after harvest, difficult to store, and its shelf life is short, which greatly limits the development of the mulberry industry. Therefore, dry processing of mulberry fruit is an important way to reduce fresh fruit loss and increase product value. There are relatively many drying methods available, but drying and direct drying are

Science and Technology Research Projects of Colleges and Universities in Hebei Province (QN2019052) Funders play a certain role in research design, data collection and analysis, publication decision or manuscript preparation.

**Competing interests:** There is no competition in this article.

usually used for mulberry fruit drying at present, both of which have advantages and disadvantages. Therefore, modern high-tech food processing technology is applied to dry mulberry, product production to extend the shelf life of food [13]. It is of great significance to improve the processing level of dried mulberry products and promote the industrialization of mulberry.

An important development trend of modern food processing technology is to maintain the nutrition, color, and flavor of food to the maximum extent. Common mulberry products include mulberry wine, dried mulberry fruit, mulberry drinks, etc. Dry processing is one of the main processing forms of mulberry, which can greatly solve the problem of intolerant storage of mulberry, and the selection of drying technology and equipment has a great influence on the nutritional quality of dried products. Hot air drying is the most commonly used drying method, but the drying speed is slower and the drying time is longer [14]. Vacuum freeze-drying is a process of removing water or other solvents from frozen biological products through sublimation. It is a raw material drying method for obtaining high quality biological products, which has the characteristics of maintaining the original biological activity, color and shape, fast sample dissolution rate and low residual moisture [15].

A large number of experimental studies have been conducted on the drying characteristics of mulberry under various drying methods at home and abroad. Mahmood *et al.* [16] used natural drying and sun drying treatments for different varieties of mulberry, and the results showed that drying conditions had significant effects on the polyphenol and antioxidant properties of mulberry. The research results of Ester *et al.* [17] showed that freeze-drying treatment made samples have better texture and color. Previous studies on hot air drying and vacuum freeze-drying of mulberry have been conducted, but there is a lack of systematic studies on the effects of hot air drying and vacuum freeze-drying on the comprehensive quality of different varieties of mulberry [18]. In this study, hot air drying and vacuum freeze-drying techniques were used to explore the effects of drying methods on the physical properties, nutritional quality, functional components and antioxidant activity of dried D1, D2 and D3 of the three mulberry varieties, which were all planted in Dingzhou for 5 years, and D1 was a white variety. D2 is a northern black variety and D3 is a southern purple black variety. To provide a theoretical reference for choosing different mulberry drying treatment methods, the changes in dried fruit quality were discussed.

## 2 Materials and methods

Mulberries (D1, D2 and D3) were all picked in Dongsheng Ecological Park, Dingzhou City, Hebei Province. Mulberry was picked manually and each sample was put into a separate bag after picking. The mulberry was tested immediately after being transported to the laboratory.

Vacuum freeze-drying and hot air drying are adopted.

The chemical reagents used in the whole study were of analytical grade. 2, 2'-Azino-bis (3-ethylbenzothiazoline-6-sulfonic acid), 2,2-diphenyl-1-picrylhydrazyl (EPR spectroscopy), 2,4,6-tris(2-pyridyl)-s-triazine and Folin-Ciocalteu were purchased from Solarbio, and bovine serum protein, glycine, rutin, resveratrol and chlorogenic acid standards were purchased from Sourleaf Biological Co., Ltd. Reagents such as anthrone, concentrated sulfuric acid, ninhydrin, sodium hydroxide, ethanol, sodium carbonate and sucrose were purchased from Sinopharm Chemical Reagent Co., Ltd.

### 2.1 Drying process

The fresh mulberry was dried in a vacuum freeze dryer and air drying oven. Two different drying methods were used to take a fixed weight sample (150 g). All fresh mulberries were dried

until they reached a constant weight. The following drying modes were used for sample preparation. All fresh mulberry fruits were dried until they reached a constant weight. The following drying mode was used for the preparation of samples.

**2.1.1 Vacuum freeze-drying.** Fresh mulberry (150 g) was completely frozen in a deep refrigerator at -50°C (24 hours) and then dried. Then, a vacuum freeze-dryer (FDU-1200) was used to dry for 48 h in a vacuum freeze-dryer under 20 pA at a constant temperature of -50°C.

**2.1.2 Hot air drying.** Fresh mulberry was spread on a perforated steel plate and dried in hot air at 60°C for 48 h.

## 2.2 Physical and chemical analysis

Wettability was measured by referring to the method of Jumah et al. [19]. Then, 100 mL distilled water was added to a 250 mL beaker, 0.5 g mulberry powder was evenly spread on the water surface, and the time (s) required for all the powder to be wetted by water was recorded immediately. The test was repeated three times, and the average value was taken to represent the wettability of dried mulberry powder.

The moisture absorption method of a reference person such as Zhao [20] takes mulberry 3 g dried fruit powder, evenly spread in clean in a petri dish, and then puts the glass dish in an enamel disc, adding enough and saturated sodium chloride solution, after the two layers of sealing surface with plastic wrap and place for 7 days, accurate said in petri dishes and according to the formula to calculate the quality with the sample moisture absorption. The formula is as follows:

$$\text{Hygroscopicity (\%)} = \frac{\frac{m}{M+m} \times 100}{1 + m/M}$$

where m is the weight change of mulberry powder before and after (g).

M: initial mass of mulberry powder (g).

The method of Hameed et al. [21] was used for determination by hydraulic reference. Mulberry leaf powder (0.5 g) was accurately weighed into a 10 mL centrifuge tube, 10 mL distilled water was added, thoroughly mixed for 10 min, and then centrifuged at 5000 R/min for 10 min. The supernatant was discarded, the residual water on the wall of the centrifuge tube was dried with filter paper, and the mass was weighed. This was repeated three times, and the average value and standard deviation were used for drawing. The formula is as follows:

$$\text{Hydraulic (g/g)} = (M - m)/m$$

where M is the mass of dried fruit powder after centrifugation (g).

m: Initial mass of dried fruit powder (g).

The bulk density was measured according to the method of Sowbhagya et al. [22]. A certain amount of mulberry powder was put into a 10 mL centrifuge tube and shocked until the mulberry powder was just filled to the calibration line of the volumetric bottle. The bulk density (DO) of the powder was calculated by the following formula:

$$\text{do (g/m L)} = (M - m)/10$$

where M is the total weight of the fruit powder and centrifuge tube (g).

m: Test tube weight (g).

According to the method of Hameed et al. [21], 0.2 g of mulberry powder was placed in a centrifuge tube, 10 mL of distilled water was added, and the mixture was stirred evenly. After centrifugation at 5000 R/min for 5 min, 5 mL of the obtained supernatant was placed into a glass Petri dish, which was placed in an oven, and dried at 105°C for 3 h, and the total mass

after drying was weighed. The formula is as follows:

$$\text{solubility } (\%) = (M - m) \times 2 \times 100/m_1$$

where M is the total weight of the dried Petri dish and dissolved fruit powder (g).

m: Net weight of Petri dish (g).

$m_1$: Initial weight of mulberry powder (g).

## 2.3 Nutrient content

Soluble protein was determined according to the method of Bradford [23]. The mulberry sample (0.1 g) was accurately weighed and placed in a mortar with 5.0 mL $H_3PO_4$ buffer (pH = 7.8), and fully ground. Then the mortar was rinsed with 5.0 mL of the above buffer and poured into a 10.0 mL centrifuge tube. The samples were centrifuged at a constant temperature of 10000 RMPs for 20 min at 4°C. Then, 1.0 mL of the supernatant was aspirated, 5.0 mL of Coomassie brilliant blue G-250 solution was added, and the absorbance value was measured at 595 nm after 2 min of mixing.

The soluble sugar content was determined according to the method of Candida [24]. The mulberry sample of 0.1 g was accurately weighed and ground thoroughly with 10.0 mL distilled water in a mortar, placed in a 10 mL centrifuge tube, sealed and heated in a boiling water bath for 30 min. The heated solution was filtered through gauze, the rinsed residue was filtered repeatedly with distilled water, and the volume was stabilized into a 25 mL volumetric flask. Then, 1.5 mL of distilled water, 0.5 mL of anthrone, ethyl acetate reagent and 5.0 mL of concentrated sulfuric acid were added to the test tube in sequence and mixed with shock. The absorbance value was measured at 630 nm wavelength after the sample was cooled to room temperature. Finally, the soluble sugar content of the samples was calculated according to the standard curve and formula.

The content of free amino acids was determined by reference to the ninhydrin chromogenic method [25]. First, the mulberry sample was weighed by an analytical balance and ground into a slurry in a mortar. Then, 5.0 mL distilled water was added to the mortar and placed into the centrifuge tube. Then 5.0 mL distilled water was added to rinse the mortar and poured into the centrifuge tube. Ultrasonic extraction was carried out at 40°C for 20 min using an ultrasonic cleaning machine. After extraction, it was centrifuged at 5000 rpm for 25 min, and 1.0 mL of the supernatant was absorbed by a pipetting nozzle, Then, 2.0 mL of $H_3PO_4$ buffer with pH 6.0 was added, and 3.0 mL of ninhydrin solution was added after full shaking and standing for 5 min. The centrifuge tube was sealed and heated in a boiling water bath for 30 min. After cooling, the absorbance was measured at 560 nm, and the amino acid content was calculated according to the regression equation.

Determination of ascorbic acid (VC) content: The method of Habib *et al.* [26] was used for determination.

## 2.4 Content of functional components

The content of flavonoids in mulberry was determined by the $AlCl_3$ chromogenic method [27]. First, a 0.2 g mulberry sample was accurately weighed by an analytical balance, and 5.0 mL 70% $C_2H_5OH$ was added according to a material to liquid ratio of 1:25, and placed into a centrifuge tube. Then, an ultrasonic cleaning machine was used for extraction for 40 min at a constant temperature of 25°C. The extracted extract was centrifuged at 7000 RPM for 10 min. For determination, 1.0 mL of supernatant was taken, 0.4 mL of 50% $NaNO_3$ solution and 0.4 mL of 10% $AlCl_3$ solution were added, the mixture was thoroughly shaken and left for 5 min, and 4 mL of 4% NaOH solution was added. A constant volume of 70% $C_2H_5OH$ was used to

10.0 mL. After standing for 10 min at room temperature, the absorbance value was measured at a wavelength of 415 nm. Finally, the content of flavonoids in the samples was calculated according to the standard curve.

The content of polyphenols was determined by the Folin-Ciocalteu colorimetric method [28]. First, a 0.1 g mulberry sample was accurately weighed by an analytical balance, and 10.0 ml 50% $C_2H_5OH$ was added according to a solid-liquid ratio of 1:100. The samples were extracted by ultrasonication for 40 min at a constant temperature of 25°C. This was followed by centrifugation at 7000 rpm for 10 min. After centrifugation, 1.0 mL of the supernatant was taken, 1.0 mL of folinol reagent was added, thoroughly shaken and mixed, and the mixture was allowed to stand for 5 min at room temperature. Then, 4.0 ml 20% $NACO_3$ solution was added and heated in a 50°C water bath for 1 hour. After cooling to room temperature, the absorbance value was measured at 760 nm wavelength. Finally, the content of polyphenols in the samples was calculated according to the standard curve regression equation.

The resveratrol content was determined by the $C_2H_5OH$ extraction method [29]. A mulberry sample of 0.2 g was accurately weighed, and 60% $C_2H_5OH$ (3.0 mL) was added according to a solid-liquid ratio of 1:15, and placed into a 10.0 mL centrifuge tube. After shaking well, the mulberry sample was heated in a water bath at 60°C for 1 h, and the mulberry sample was kept away from light during the whole process. After filtration, samples were centrifuged at 7000 rpm for 10 min. For determination, 0.5 mL of supernatant was diluted 10 times with the above concentration of $C_2H_5OH$, the absorbance value was measured at 305 nm wavelength, and the content of resveratrol was calculated according to the standard curve and regression equation.

The chlorogenic acid content was determined by the Fe2+ chromogenic method [30]. The 0.2 g sample was accurately weighed, and 10.0 ml 60% $CH_3OH$ was added as the extraction agent according to the solid-liquid ratio of 1:5. The samples were extracted by ultrasound for 30 min. After simple filtration, 60% $CH_3OH$ was used to a constant volume of 15.0 mL, 5.0 mL of constant volume extract was taken, 0.5 mL of 0.2 mol/L $FeCl_3$ solution was added, and the mixture was shaken and left for 60 min. The absorbance value was measured at a wavelength of 755 nm, and the chlorogenic acid content was calculated according to the standard curve and regression equation.

The anthocyanin content in mulberry was determined by the pH difference method [31]. A 0.2 g mulberry sample was accurately weighed by an analytical balance, 4.0 ml of 60% acidified $C_2H_5OH$ was added according to a 1:20 solid-liquid ratio, and the whole process was kept away from light. Then the samples were placed in a 60°C water bath and heated for 2 h. The samples were allowed to cool to room temperature. Then 1.0 mL supernatant and 9.0 mL $H_3PO_4$ buffer with pH = 1.0 and pH = 4.5 were added to two 25 mL vials, shaken well, and left in the dark for 2 hours. The absorbance values were measured at wavelengths of 520 nm and 700 nm.

## 2.5 Antioxidant properties

DPPH free radical scavenging ability, ABTS free radical scavenging ability and iron reducing power of mulberry [32].

The DPPH radical scavenging ability was determined by the DPPH radical scavenging method. A mulberry sample of 0.1 g was accurately weighed by an analytical balance, and 50% $C_2H_5OH$ (0.5 mL) was added according to a solid-liquid ratio of 1:50. After ultrasonic extraction, the volume was fixed to 10.0 mL for later use. For measurement, 0.5 mL of the extract was taken, 2.0 mL of DPPH reaction solution was added, and the reaction was shaken and kept away from light for 30 min. The light absorption value was measured at a wavelength of 517 nm. In the control group, the same volume of $CH_3OH$ was used instead of the extract, and the final results were expressed as μmol/LTrolox equivalent antioxidant capacity. The

measured absorbance value was taken as the abscissa, and the concentration of Trolox standard solution was taken as the ordinate. The standard curve was drawn, and the regression equation was calculated.

In the determination of ABTS free radical scavenging ability, the method of preparing the extract was the same as that of DPPH, and ABTS+ working solution was prepared in advance:10.0 ml ABTS solution of 7 mmol/L and 5.0 potassium persulfate solution of 7.35 mmol/L were mixed, placed in the dark for 16 h at room temperature, and then prepared for use. The absorbance was diluted to 0.70 ± 0.02 at 734 nm when used. At the time of measurement, 0.5 mL of the extract was taken, 2.0 mL ABTS+ working solution was added, and the reaction was kept away from light for 6 min after shaking. The absorbance value was measured at a wavelength of 734 nm. The final results were expressed as µmol/LTrolox equivalent antioxidant capacity. The measured absorbance value is the abscissa, the concentration of Trolox standard solution is the ordinate, the standard curve is drawn, and the regression equation is calculated.

Iron reducing capacity was determined by the FRAP method. The method of preparing the extract in the determination is the same as that in DPPH. Before the test, the TPTZ working solution was prepared in advance: $CH_3COONa$ solution pH = 3.6. Then, 300 mmol/L, 10 mmol/L, TPTZ solution, and 20 mmol/L, $FeCL_3 \cdot 6H_2O$ solution were mixed according to a volume ratio of 1:1:1. For measurement, 0.5 mL of extract was taken, 1.0 mL of distilled water and 1.8 mL of TPTZ working solution were added, and the reaction was kept away from light for 10 min. The absorbance value was measured at a wavelength of 593 nm, and the final result was expressed as the antioxidant capacity of µmol/L Trolox. The measured absorbance value is the abscissa, and the concentration of Trolox standard solution is the ordinate. The standard curve was drawn and the regression equation was calculated.

## 2.6 Statistical analysis

A completely randomized design (CRD) was used to analyze the physicochemical properties, nutrients, functional components and antioxidant properties of mulberry fruit. One-way analysis of variance ($P <0.05$) was used to analyze significant differences between treatments. The parameters of chemical and antioxidant properties of different varieties of mulberry were repeated three times in these studies.

## 3 Results and discussion

Table 1 shows the physical properties of mulberry fruit under different drying modes (vacuum freeze-drying and hot-air drying). Fig 1 shows the changes in the nutrient composition of mulberry fruit under different drying modes. Fig 2 shows the changes in the functional components of mulberry fruit under different drying modes. Fig 3 shows the variation in the antioxidant capacity of mulberry fruit under different drying modes.

## 3.1 Changes in physical characteristics under different drying treatments

Table 1 describes the effects of the different drying processes used in this study on fruit wettability, hygroscopicity, water retention, bulk density and solubility. In terms of physical and chemical properties, fruit wettability, moisture absorption, water retention, bulk density and solubility all reflect the effects of drying processes on fruit quality. The results in Table 1 show that the physicochemical properties of different mulberry varieties after vacuum freeze-drying were better than those after hot air drying, which indicated that the use of exhaust heat from the condenser as the heat source could significantly accelerate the rate of water content decline.

**Table 1. Effects of different drying methods on the physical properties of mulberry fruit powder.**

| Varieties | process mode | Wettability(s) | Hygroscopicity(%) | water-holding power(g/g) | bulk density(g/mL) | Solubleness(%) |
|---|---|---|---|---|---|---|
| D1 | vacuum freeze drying | 241.00±22.07aA | 9.13±1.59aA | 2.09±0.05aA | 0.46±0.07aA | 28.15±14.68aA |
| | hot air drying | 55.00±8.50bB | 4.69±2.90bB | 1.69±0.33aA | 0.53±0.14bB | 35.03±0.84aA |
| D2 | vacuum freeze drying | 65.00±7.23aA | 10.41±0.47aA | 2.09±0.09aA | 0.54±0.09aA | 38.41±2.50aA |
| | hot air drying | 27.00±5.86bB | 7.74±0.72bB | 1.38±0.02bB | 0.52±0.07aA | 41.57±2.41aA |
| D3 | vacuum freeze drying | 144.00±20.60aA | 10.35±0.68aA | 2.17±0.01aA | 0.47±0.21aA | 37.75±0.81aA |
| | hot air drying | 110.00±10.50aA | 7.39±0.82bB | 1.24±0.00bB | 0.66±0.59bB | 43.48±9.74aA |

Note: Lowercase letters indicate the significance of differences between different treatments of the same variety ($P<0.05$), uppercase letters indicate the significance of differences between different treatments of the same variety ($P<0.01$).

These data indicate that an important feature of vacuum freeze dryers is the efficient use of energy to reduce moisture content compared to other drying methods.

Moisture content is a very important characteristic of freeze-dried foods, as it affects the appearance, texture and taste of the product. Moisture content also affects the freshness and shelf life of products. A high moisture content will make it easy for bacteria, mold and yeast to multiply, and the food will change. The vacuum freeze-drying process removes moisture from the sample. In terms of the physical properties of mulberry, the fruit powder with packing density, hygroscopic property, wettability and solubility after vacuum freezing is better than that after hot air drying, which is related to the hardening phenomenon on the surface of the mulberry powder after hot air drying and the inability to form a large number of micropores inside the fruit powder, which is consistent with the results of Caparino [33].

## 3.2 Changes in nutrient content under different drying treatments

The results of Fig 1 show that after vacuum freeze-drying and hot-air drying, the nutrient content of mulberry fruits of the three varieties changed. Compared with fresh mulberry fruit, the content of mulberry fruit showed a decreasing trend after different drying treatments, but the retention of these components by vacuum freeze-drying was better than that by hot air drying, which may be because hot air drying would promote the Maillard reaction and Strecker degradation reaction of amino acids in mulberry fruit, resulting in the loss of amino acids [34]. The reason why vacuum freeze-drying reduces the content of amino acids may be because freezing can prevent proteins from degrading into amino acids [35]. The research results of Paramanandam *et al.* [36] found that the content of amino acids and proteins in fruits under hot air drying was higher, which is different from the research results in this paper and needs to be further verified. In addition, the research results of Harguindeguy and Fissore [37] show that different drying processes have an impact on the ascorbic acid content, phenolic compounds and antioxidant properties of food.

## 3.3 Content changes of functional components under different drying treatments

Fig 2 shows the changes in the functional components of the three mulberry fruits under different drying modes. According to the data, compared with fresh mulberry fruit, the contents of total phenols, total flavonoids, ascorbic acid and neo-chlorogenic acid under vacuum freeze-drying were significantly higher than those under hot air drying after different drying treatments. This was because the functional components such as polyphenols, flavonoids and vitamins in mulberry were affected by light, temperature, oxygen partial pressure, water

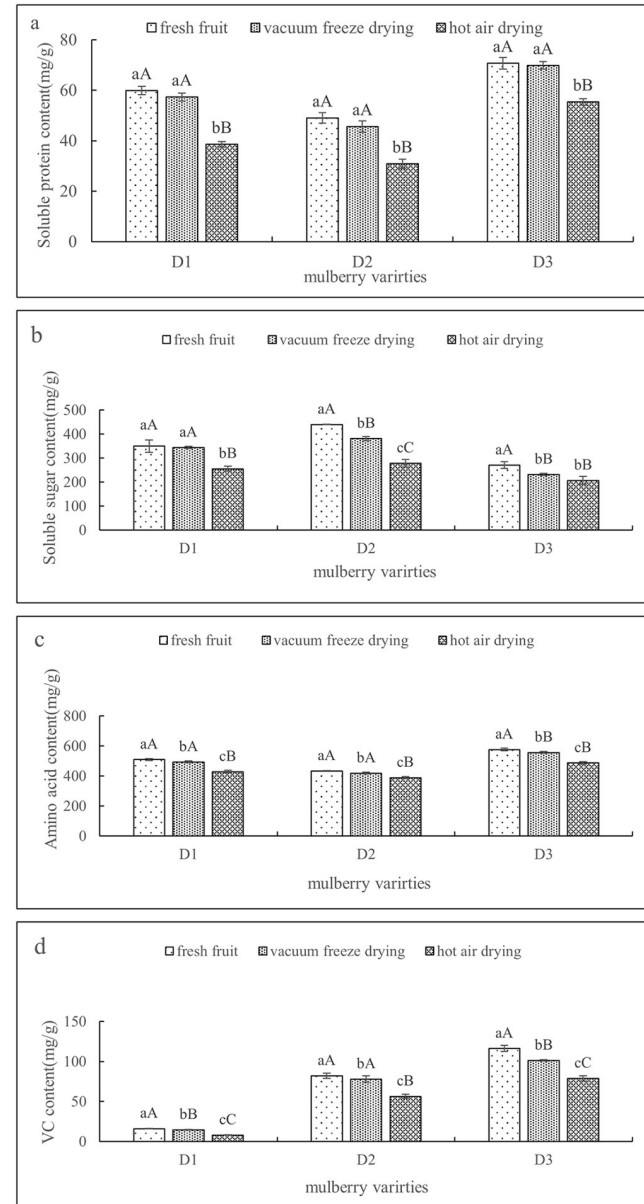

**Fig 1. (a, b, c, d) shows the changes in soluble protein, soluble sugar, amino acid and ascorbic acid contents of fruits under different drying treatments (error bars indicate the standard error of the average).** Different letters a-c in the same column indicate significant differences ($P < 0.05$).

activity and other factors during the drying process, and oxidation, aggregation or decomposition occurred, resulting in changes in the content of functional components [38–40]. In addition, under the conditions of low temperature and low oxygen partial pressure, the activity of oxidase is low. Therefore, thermally sensitive components such as polyphenols, flavonoids and ascorbic acid easily undergo enzymatic oxidation and have better retention under this dry condition. Hot air drying destroys heat sensitive components, such as flavonoids [41, 42]. The research results of Angela *et al.* [43] showed that compared with hot-air drying, the content of flavonoids and phenolic acids was higher under freeze-drying. Tan *et al.* [44] found that the

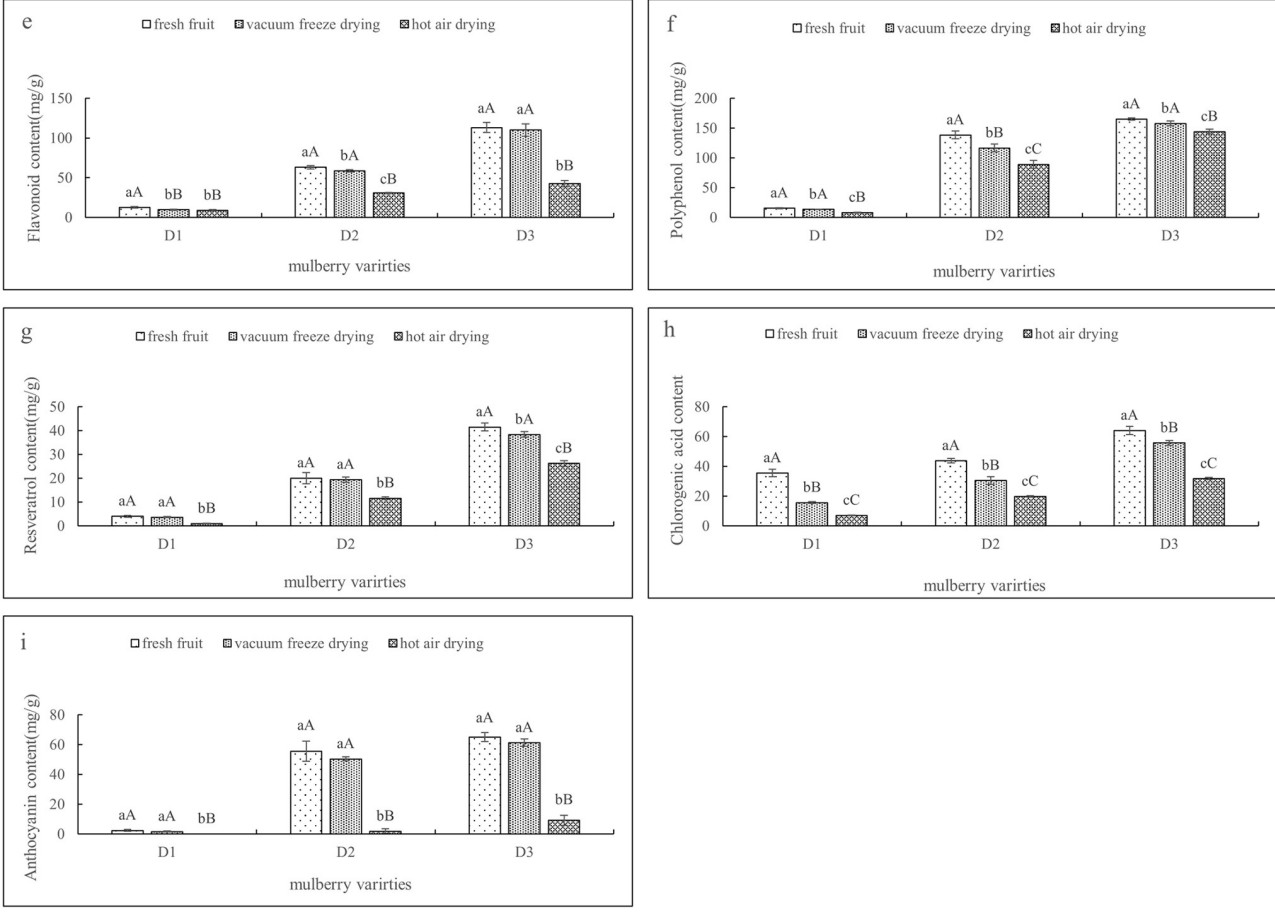

**Fig 2. (e, f, g, h, i) shows changes in the contents of flavonoids, polyphenols, resveratrol, chlorogenic acid and anthocyanins in fruits under different drying treatments (error bars indicate the standard error of the average).** Different letters a-c in the same column indicate significant differences ($P < 0.05$).

content of anthocyanins in fresh fruits was the highest and decreased with increasing drying temperature. According to the analysis, cyanidin-3-O-glucoside and delphinidin-3-O-galactoside were the main anthocyanins in flesh-and-blood peaches, but during the whole drying process, the content of delphinidin-3-O-galactoside decreased continuously, which was consistent with the results of this study.

## 3.4 Changes in antioxidant properties under different drying treatments

Total phenols and flavonoids were the main functional components of mulberry, and phenolic compounds composed of chlorogenic acid and its derivatives were dominant. Polyphenol compounds contain multiple phenolic hydroxyl groups and have strong reducing properties. Their structural properties are the main reason for the antioxidant activity of these compounds. They achieve antioxidant activities mainly by directly scavenging free radicals or interacting with metal ions. The results showed that the antioxidant activity of mulberry under vacuum freeze-drying was stronger, which might be due to the better retention and reducibility of phenolic compounds under vacuum freeze-drying. Ozkan *et al.* [45] reported a similar trend in their findings. This was also confirmed by the results of Zhang *et al.* [46], who found that the scavenging activity of chlorogenic acid on DPPH and ABTS free radicals reached

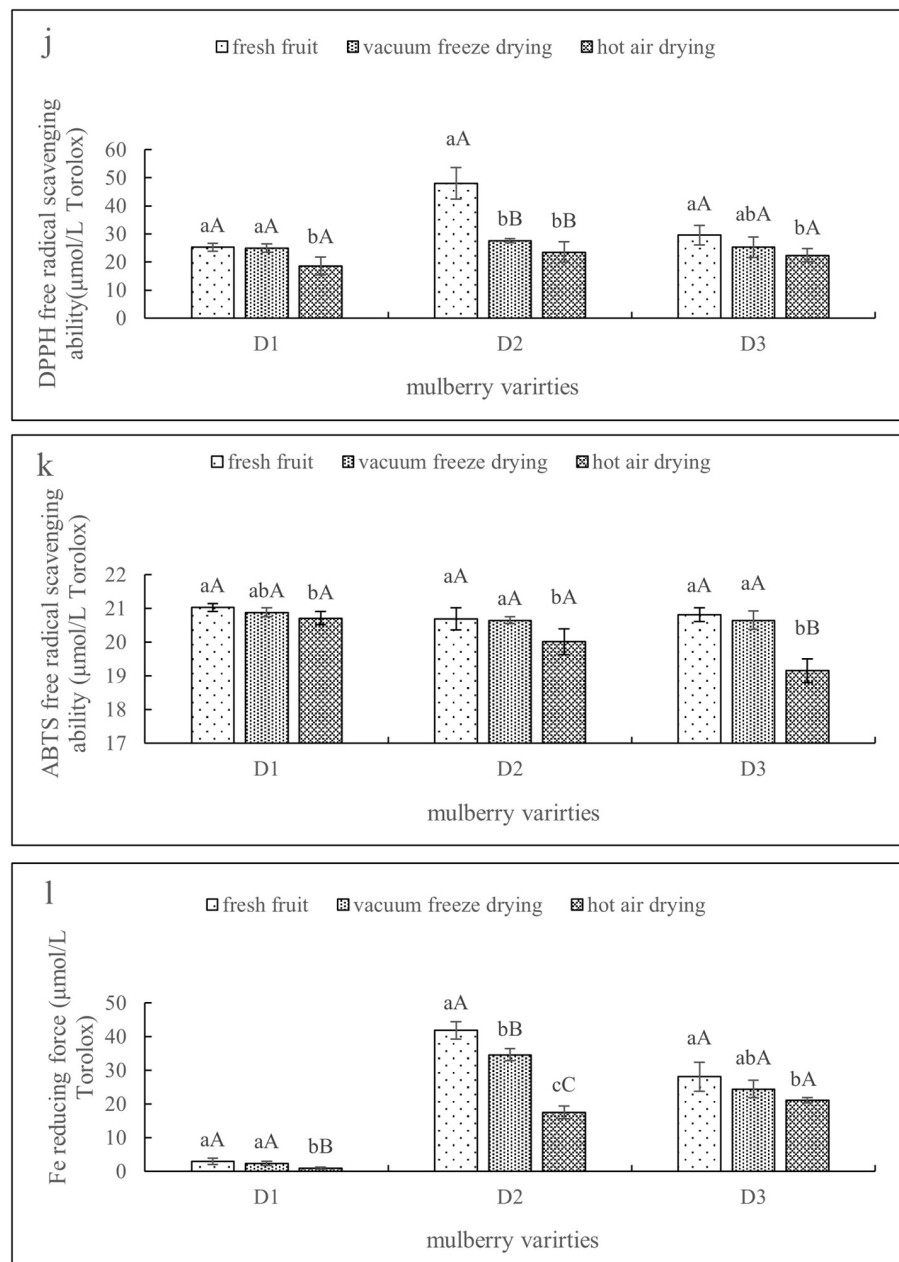

**Fig 3. (j, k, l) shows the changes in DPPH free radical scavenging ability, ABTS free radical scavenging ability and iron reducing power of fruits under different drying treatments (error bars indicate the standard error of the average).** Different letters a-c in the same column indicate significant differences ($P < 0.05$).

92.4% and 99.4% respectively. Chen *et al.* [47] determined the total phenols, flavonoids, vitamin C and antioxidant capacity of 25 mulberry samples from 6 provinces, and the results showed that D3 had the highest health care value and the highest active compound content. The data in the same figure show that the antioxidant activity of mulberry under vacuum freeze-drying is high, while that under hot-air drying is low. Therefore, the contents of total phenols, flavonoids and antioxidant substances in mulberry have great influence on the antioxidant capacity of mulberry, and the contents of functional components in mulberry fruits,

especially phenolic substances, are affected by different drying methods. The drying method also determines its oxidation resistance.

In conclusion, both hot air drying and vacuum freeze-drying had significant effects on the dry quality characteristics of mulberry fruit, and vacuum freeze-drying could more effectively retain the nutritional quality, functional components and antioxidant activity of fresh mulberry fruit. This study provides an effective reference for the further development and utilization of mulberry products. It can also provide some reference for further study on the effect of drying methods on mulberry fruit drying. Abbaspour, Igor and Duan *et al.* [48–50] stated that freeze-drying has the lowest color and shrinkage rate of fruits, the highest content of bioactive compounds, and better rehydration performance, and can maintain the original quality to the maximum extent, so it has broader application prospects in food, medicine, biological products and other fields. However, the relationship between the drying method and fruit characteristics should be considered at the same time. Ai *et al.* [51] found in their research results on the fruit of Amomum chinensis that freeze-drying can make the fruit of Amomum chinensis have a complete glandular hair structure, which has the best color retention effect, the lowest shell breaking rate and the best flavor profile retention, but it has the longest drying time and the highest energy consumption. Therefore, in the actual production process, attention should be given to the relationship between productivity and income, and the most suitable treatment method for mulberry drying should be selected on the basis of minimizing the economic cost.

## 4 Conclusion

In this study, it is believed that mulberry fruits are rich in nutrients, functional components and antioxidant properties, and the vacuum freeze-drying method is higher than the hot air drying method in nutrients such as amino acid content, functional components such as polyphenols, flavonoids and antioxidant properties such as DPPH free radical scavenging ability of mulberry fruits. However, different drying methods had different effects on mulberry fruits. The contents of various indexes of dried mulberry fruits after vacuum freeze-drying and hot air drying decreased significantly compared with fresh mulberry fruits, but the loss rate of vacuum freeze-drying was the lowest, the loss rate of hot air drying was higher, and the shape and color of mulberry fruits could be better preserved by freeze-drying. Vacuum freeze-drying treatment can preserve the active ingredients in mulberry to the greatest extent and is suitable for processing various nutritional foods in the future.

## Supporting information

**S1 File.**
(DOCX)

## Author Contributions

**Investigation:** Ningwei Yang.

**Methodology:** Haichao Wen.

**Writing – original draft:** Li Wang.

**Writing – review & editing:** Hongjiao Li.

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
