## [Decision Letter · Decision Letter 0]

31 Jan 2023

PONE-D-22-29075Effect of vacuum freeze drying and hot air drying on dried mulberry fruit qualityPLOS ONE

Dear Dr. Hongjiao Li,

Thank you for submitting your manuscript to PLOS ONE. After careful consideration, we feel that it has merit but does not fully meet PLOS ONE’s publication criteria as it currently stands. Therefore, we invite you to submit a revised version of the manuscript that addresses the points raised during the review process.

We look forward to receiving your revised manuscript.

Kind regards,

Awatif Abid Al-Judaibi, PhD

Academic Editor

PLOS ONE

Journal Requirements:

Reviewers' comments:

Reviewer's Responses to Questions

**Comments to the Author**

1. Is the manuscript technically sound, and do the data support the conclusions?

Reviewer #1: Yes

Reviewer #2: Yes

2. Has the statistical analysis been performed appropriately and rigorously? 

Reviewer #1: Yes

Reviewer #2: Yes

3. Have the authors made all data underlying the findings in their manuscript fully available?

Reviewer #1: Yes

Reviewer #2: Yes

4. Is the manuscript presented in an intelligible fashion and written in standard English?

Reviewer #1: No

Reviewer #2: No

5. Review Comments to the Author

Reviewer #1: Review Report

Thank you for giving me the opportunity to review your manuscript entitled” Effect of vacuum freeze drying and hot air drying on dried mulberry fruit quality”. I very much enjoyed reading your manuscript. The authors propose that vacuum freeze drying is better method of mulberry fruit drying than hot air drying. They have conducted various test and illustrated the results an appropriate way, with most components in the mulberry fruits using vacuum freeze drying approach are significantly high. Conversely, the findings are seems not novel and previously investigated in few studies.

Major comments:

I. The findings are not novel and previous studies have already demonstrated that vacuum freeze drying is a better method of drying, but the study has conducted in a very systematic way and has contribution to the mulberry fruit drying.

II. The introduction of the manuscript is not adequately structured and need more information to be added.

III. The material methods are mentioned in detail, but there are some repetitions, which have compromised the quality of the methodology. In addition the authors should write materials and methods comprehensively.

IV. The data is properly analyzed and have presented in well-illustrated figures, but results need more appropriate description and discussion, the authors should highlight the important findings.

Minor Comments:

i. In the abstract line 5 to 8 are weird, the authors need to rephrase the sentence.

ii. In Introduction line 4, the authors should paraphrase the sentence to give correct sense.

iii. In Intro line 8 and 9 the sentence making no sense they should be rephrased.

iv. The format of manuscript should be carefully rechecked to correct mistakes in the format.

v. English must go through a revision, there are some grammar mistakes that compromise the article.

Overall your manuscript has potentials and dealing with the interesting topic. The hypotheses seems obvious and the methods simplistic. The vacuum freeze drying is a better method than the hot air drying. The work has important results, but there are a few doubts listed above that must be clarified before accepting for publication. Therefore I recommend for publication only if these doubts and questions are totally clarified and major changes are done.

Reviewer #2: General comment: -

In general, the paper needs to be written properly with good international English and grammar to improve the quality of the English language. The authors must avoid silly mistakes and other typographical errors. The authors should read the paper multiple times to avoid simple and common mistakes. I would suggest the authors to check the plagiarism (please follow the journal rules!) after writing the paper to avoid any future problem.

Specific Comments:

1. The paper is not written well scientifically (please see below!). The authors should maintain a standard English all through the paper taking care of the grammar.

2. The authors should write a more proper introduction with proper literature review (information) and references.

3. The authors should mention the logic of using the given 3 cultivars of the mulberry fruit and should mention how they are different among themselves (it must be written in the introduction).

4. Please write the references for all the information mentioned in the paper e,g ‘’Previous studies on hot air drying and vacuum freeze-drying of mulberry were carried out, but there was a lack of systematic study on the effects of hot air drying and vacuum freeze-drying on the comprehensive quality of different varieties of mulberry…..’’

Here the references are missing ??

5. Please explain the ‘’Methods Section’’ properly and in detail. Vital information related to method is/are missing. E.g., What is the processing time taken of the fruits from the field to the actual processing of the samples (for the experiments)?

6. Please write the full name of all the chemical used at the place of its first occurrence in the manuscript and later use the abbreviations e.g ABTS, DPPH, TPTZ. Authors have not mentioned the detail of the chemicals used.

7. Authors should clarify why they used different ‘’anti-oxidant assays’’ to measure the same thing? Please mention the logic and the benefit if any (also mention references)?

8. In ‘’Materials and Method section’’ the authors have mentioned ‘’Mulberry (Baiyuwang (D1), Longsang (D2) and Zhongshen 1 (D3)) were picked from Dongsheng Ecological Park in Dingzhou City,….’’ The author should use the code D1, D2 and D3 in the abstract itself where they have mentioned the name of the different cultivar of the mulberry. After that the authors should stick to the code (no need to mention full name everywhere!).

9. In the manuscript, the authors have mainly written the results!! They must discuss their results (Discussion part) in proper and a better logical way based on their findings (results) and the known literature in this topic. The results and the discussion part must be well written and should make a better sense.

10. The ‘’Conclusion Section’’ should also be ‘’re-written’’ properly based on the results with future perspective etc.

6. PLOS authors have the option to publish the peer review history of their article (what does this mean?). If published, this will include your full peer review and any attached files.

Reviewer #1: No

Reviewer #2: **Yes: **Dr. Arif Mohammed

---

## [Author Response · Author response to Decision Letter 0]

14 Feb 2023

Reply to all comments point-by-point (Manuscript Number: PONE-D-22-29075)

The authors’ replies are in blue.

COMMENTS FOR THE AUTHOR:

Reviewer 1#

1.The findings are not novel and previous studies have already demonstrated that vacuum freeze drying is a better method of drying, but the study has conducted in a very systematic way and has contribution to the mulberry fruit drying.

[Reply] Thank you very much for your comments. Yes, there have been many researches on fruit drying methods, but most of them focus on the influence of drying methods on fruit quality, or the advantages and disadvantages of different drying methods. This paper mainly studies the physical and chemical analysis, nutritional composition, functional composition and antioxidant capacity of fruits under different drying methods. This is one of the reasons for our research. We hope to use mulberry as the test material to get the difference of fruit quality under different drying methods.

2 The introduction of the manuscript is not adequately structured and need more information to be added.

[Reply] The introduction of this article has been added. Thank you for your valuable comments.

3. The material methods are mentioned in detail, but there are some repetitions, which have compromised the quality of the methodology. In addition the authors should write materials and methods comprehensively.

[Reply] Thank you very much for your suggestion. We have rewritten the materials and methods.

4. The data is properly analyzed and have presented in well-illustrated figures, but results need more appropriate description and discussion, the authors should highlight the important findings.

[Reply] We have corrected the part of description and discussion, thank you for your suggestion.

5. In the abstract line 5 to 8 are weird, the authors need to rephrase the sentence.

[Reply] The abstract of this paper has been revised. please see line 12-19 of the “Revised Manuscript with Track Changes”, Thank you for your valuable comments.

6. In Introduction line 4, the authors should paraphrase the sentence to give correct sense.

[Reply] Thank you very much for your suggestion. We have corrected it.

7. In Intro line 8 and 9 the sentence making no sense they should be rephrased.

[Reply] Thank you very much for your suggestion. We have corrected it.

8. The format of manuscript should be carefully rechecked to correct mistakes in the format.

[Reply] We have corrected it according to the manuscript format. Thank you for your comments.

9. English must go through a revision, there are some grammar mistakes that compromise the article.

[Reply] We have modified the English language appropriately. Thank you very much for your valuable suggestions. We will continue our efforts to improve our English in the future.

COMMENTS FOR THE AUTHOR:

Reviewer 2# 

1.The paper is not written well scientifically (please see below!). The authors should maintain a standard English all through the paper taking care of the grammar.

[Reply] We have modified the English language appropriately. Thank you very much for your valuable suggestions. We will continue our efforts to improve our English in the future.

2. The authors should write a more proper introduction with proper literature review (information) and references.

[Reply] Thank you very much for your suggestion. The introduction part has been revised and appropriate references have been added. please see line 24-65 of the “Revised Manuscript with Track Changes”

3. The authors should mention the logic of using the given 3 cultivars of the mulberry fruit and should mention how they are different among themselves (it must be written in the introduction).

[Reply] Thank you very much for your suggestions. We have revised the manuscript. In this paper, three mulberry varieties were planted in Dingzhou for 5 years. Bai Yu Wang was a white variety, Longsang was a northern black variety, and Zhong Shen 1 was a southern purple black variety. please see line 62-63 of the “Revised Manuscript with Track Changes”

4. Please write the references for all the information mentioned in the paper e,g ‘’Previous studies on hot air drying and vacuum freeze-drying of mulberry were carried out, but there was a lack of systematic study on the effects of hot air drying and vacuum freeze-drying on the comprehensive quality of different varieties of mulberry…..’’

Here the references are missing ??

[Reply] We have added the references for this part, Please see line 421-423 of the “Revised Manuscript with Track Changes”.

5. Please explain the ‘’Methods Section’’ properly and in detail. Vital information related to method is/are missing. E.g., What is the processing time taken of the fruits from the field to the actual processing of the samples (for the experiments)?

[Reply] Thank you very much for your suggestion. We have added this part. Please see line 68-69 of the “Revised Manuscript with Track Changes”.

6. Please write the full name of all the chemical used at the place of its first occurrence in the manuscript and later use the abbreviations e.g ABTS, DPPH, TPTZ. Authors have not mentioned the detail of the chemicals used.

[Reply] We have made some changes in the article, Please see line 71-73 of the “Revised Manuscript with Track Changes”. Thank you very much for your valuable advice.

7. Authors should clarify why they used different ‘’anti-oxidant assays’’ to measure the same thing? Please mention the logic and the benefit if any (also mention references)?

[Reply] References have been reviewed and cited, please refer to lines 489-490 of "Revised Manuscript with Track Changes".

8. In ‘’Materials and Method section’’ the authors have mentioned ‘’Mulberry (Baiyuwang (D1), Longsang (D2) and Zhongshen 1 (D3)) were picked from Dongsheng Ecological Park in Dingzhou City,….’’ The author should use the code D1, D2 and D3 in the abstract itself where they have mentioned the name of the different cultivar of the mulberry. After that the authors should stick to the code (no need to mention full name everywhere!).

[Reply] Thank you very much for your suggestions. We have used codes D1, D2 and D3 in the abstract, and will stick with them for the rest of the article.

9. In the manuscript, the authors have mainly written the results!! They must discuss their results (Discussion part) in proper and a better logical way based on their findings (results) and the known literature in this topic. The results and the discussion part must be well written and should make a better sense.

[Reply] Thank you very much for your suggestions. We have revised the manuscript. Please refer to the manuscript.

10.The ‘’Conclusion Section’’ should also be ‘’re-written’’ properly based on the results with future perspective etc.

[Reply] Thank you very much for your suggestion. We have corrected the content of the conclusion, please refer to the manuscript.

Once again, thank you very much for your comments and suggestions. Those comments are all valuable and very helpful for revising and improving our manuscript.

---

## [Editor Report · Decision Letter 1]

16 Feb 2023

PONE-D-22-29075R1Effect of vacuum freeze drying and hot air drying on dried mulberry fruit qualityPLOS ONE

Dear Dr. Hongjiao Li,

Thank you for submitting your manuscript to PLOS ONE. After careful consideration, we feel that it has merit but does not fully meet PLOS ONE’s publication criteria as it currently stands. Therefore, we invite you to submit a revised version of the manuscript that addresses the points raised during the review process.

We look forward to receiving your revised manuscript.

Kind regards,

Awatif Abid Al-Judaibi, PhD

Academic Editor

PLOS ONE

Journal Requirements:

Additional Editor Comments (if provided):

Dear Authors,

Please check the comments in the attached file.

---

## [Author Response · Author response to Decision Letter 1]

2 Mar 2023

Dear editor,

Thank you very much for sending the reviewers’ comments on our manuscript entitled “Effect of vacuum freeze drying and hot air drying on dried mulberry fruit quality” (Manuscript Number: PONE-D-22-29075). Those comments are all valuable and very helpful for revising and improving our manuscript. We have studied the comments very carefully and have made corrections for the whole paper.

We believe that we substantially revised the manuscript following the reviewers’ comments. All comments have been responded to point-to-point in “Response to Reviewers”, which also have been incorporated to our manuscript in “Revised Manuscript with Track Changes”. And the final clean version is “Manuscript”. We have also modified the picture format in “Picture”.

In addition, we have made corrections according to the format of the references in this publication. Thank you for you giving us the opportunity to submit our revised manuscript. We look forward to hearing from you soon. Best regards, 

Sincerely yours, Hongjiao Li

---

## [Editor Report · Decision Letter 2]

6 Mar 2023

Effect of vacuum freeze drying and hot air drying on dried mulberry fruit quality

PONE-D-22-29075R2

Dear Dr. Hongjiao Li,

We’re pleased to inform you that your manuscript has been judged scientifically suitable for publication and will be formally accepted for publication once it meets all outstanding technical requirements.

Kind regards,

Awatif Abid Al-Judaibi, PhD

Academic Editor

PLOS ONE

---

## [Editor Report · Acceptance letter]

12 Mar 2023

PONE-D-22-29075R2 

Effect of vacuum freeze drying and hot air drying on dried mulberry fruit quality 

Dear Dr. Li:

I'm pleased to inform you that your manuscript has been deemed suitable for publication in PLOS ONE. Congratulations! Your manuscript is now with our production department. 

Kind regards, 

on behalf of

Professor Awatif Abid Al-Judaibi 

Academic Editor

PLOS ONE